# SPARC: Scenario Planning and Reasoning for Automated C Unit Test Generation

## Abstract

Automated unit test generation for C remains a formidable challenge due to the semantic gap between high-level program intent and the rigid syntactic constraints of pointer arithmetic and manual memory management. While Large Language Models (LLMs) exhibit strong generative capabilities, direct intent-to-code synthesis frequently suffers from the leap-to-code failure mode, where models prematurely emit code without grounding in program structure, constraints, and semantics. This will result in non-compilable tests, hallucinated function signatures, low branch coverage, and semantically irrelevant assertions that cannot properly capture bugs. We introduce SPARC, a neuro-symbolic, *scenario-based* framework that bridges this gap through four stages: (1) Control Flow Graph (CFG) analysis, (2) an Operation Map that grounds LLM reasoning in validated utility helpers, (3) Path-targeted test synthesis, and (4) an iterative, self-correction validation loop using compiler and runtime feedback. We evaluate SPARC on 59 real-world and algorithmic subjects, where it outperforms the vanilla prompt generation baseline by 31.36% in line coverage, 26.01% in branch coverage, and 20.78% in mutation score, matching or exceeding the symbolic execution tool KLEE on complex subjects. SPARC retains 94.3% of tests through iterative repair and produces code with significantly higher developer-rated readability and maintainability. By aligning LLM reasoning with program structure, SPARC provides a scalable path for industrial-grade testing of legacy C codebases.

## CCS Concepts

• **Software and its engineering** → **Software testing and debugging**; *Software maintenance tools*; • **Computing methodologies** → *Natural language processing*.

## Keywords

Automated Test Generation, Large Language Models, C Unit Testing, Control Flow Analysis, Mutation Testing

**ACM Reference Format:**
Anonymous Author(s). 2018. SPARC: Scenario Planning and Reasoning for Automated C Unit Test Generation. In *Proceedings of Make sure to enter the correct conference title from your rights confirmation email (Conference acronym 'XX)*. ACM, New York, NY, USA, 9 pages. https://doi.org/XXXXXXX.XXXXXXX

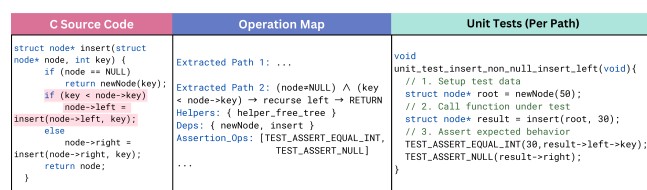

**Figure 1: An extracted execution path and its corresponding generated unit test for a BST `insert` function.**

## 1 Introduction

A unit test examines whether a specific function behaves correctly for a given set of inputs and corresponding expected outputs (Figure 1). These tests provide a foundation for refactoring, debugging, and software maintenance. Ensuring software correctness through testing in legacy C code is notoriously difficult due to pointer arithmetic, manual memory management, and complex control flow. Furthermore, in large-scale C projects, achieving comprehensive test coverage requires significant engineering effort, specifically when the tests are authored manually.

Over the past several decades, the software engineering community has proposed different techniques to automate and scale unit test generation, including fuzzing, symbolic and concolic execution, and recently using Large Language Models (LLMs) due to their impressive capabilities in code generation tasks [1, 2, 6, 7, 25, 29, 31, 38]. LLM-based techniques have been specifically shown to outperform classic automated test generation techniques. However, existing LLM-based test generation techniques often fail to produce semantically meaningful tests [10]: they either miss edge cases, generate incorrect function calls (incorrect name or incorrect signature), produce test code that does not compile, generate invalid inputs, or produce tests with assertions that do not properly reflect the semantics of the code. Figure 2 illustrates concrete failure modes of *vanilla prompt generation*[1], for an algorithmically complex code `kohonen_som_trace.c`, using `DeepSeek V3.2` [12]:

**Undefined Path Coverage.** Vanilla-prompt-generated tests frequently plateau at "happy-path" scenarios, missing edge cases such as boundary values, null pointers, and error-handling branches [36].

**Ungrounded Dependencies.** Generated tests may reference helpers or utilities absent in the project, leading to compilation failures. This is, in fact, common due to assumptions and hallucinations about the existence of a code [36].

**Lack of Traceability.** Vanilla prompt generation often produces tests that behave like black boxes. These tests commonly use generic names (e.g., `test_update_weights`) and weak assertions that only check whether some state changed (e.g., `assert(x != 0)`) that verify state change without explaining the logic path taken. This

---

[1]Vanilla prompt includes the source code of the function under test and its surrounding file context as the intra-procedural context, instructing LLM to generate tests to cover all execution paths of function under test and achieve 100% code coverage.

**Figure 2: Three failure modes of vanilla prompt generation compared with SPARC on `kohonen_som_trace.c`.**

creates a traceability gap because unit tests are intended to serve as program comprehension artifacts and executable documentation [11], a failure in a test provides no diagnostic map.

A practical challenge underlying these limitations is the disconnect between semantic reasoning and syntactic completion. Current approaches often treat test generation as a single-step completion task, exhibiting a *leap-to-code* failure mode. This frequently results in hallucination or non-compilable code when the LLM lacks a deep understanding of the project's internal logic. When an LLM lacks a semantic map of the code, it cannot discern the inter-procedural dependencies required to exercise complex data structures. This results in tests that may achieve basic line coverage but fail to improve mutation scores because they lack useful assertions and high-entropy input sequences.

To address these limitations, we present **SPARC** (Scenario Planning And Reasoning for Automated C Unit Test Generation), a scenario-based framework which decouples the problem into two coherent tasks: first, using static analysis to derive high-level testing scenarios, and second, using these scenarios as a blueprint for context-aware test synthesis. This ensures semantically meaningful tests, featuring precise inputs and deep assertions that significantly increase both the coverage and the mutation scores. We make the following contributions in this paper:

- An automated, scenario-based framework, SPARC, that decomposes test generation into per-path scenarios and merges them into coherent unit tests (Section 3). To the best of our knowledge, SPARC is the first technique to propose the idea of scenarios in automated C test generation to bridge the semantic gap between the function under test and the test code.
- Empirical evaluation on 59 real-world and algorithmic C projects shows SPARC achieves 31.36% higher line coverage and 26.01% higher branch coverage than a vanilla prompt generation baseline, matches or outperforms KLEE on complex subjects, and retains 94.3% of generated tests through iterative repair (Section 4.1–4.2).
- A 20.78% mutation score advantage and a developer study ($n$=10) confirm superior fault detection and higher ratings in readability, correctness, completeness, and maintainability. (Section 4.2–4.4)
- Scalability and LLM-portability analysis demonstrates that cost-efficient models match frontier performance, showing that the pipeline architecture drives test quality (Section 4.5–4.6).

## 2 Related Works

**Classical Automated C Testing.** Traditional automated testing for C employs symbolic/concolic execution (e.g., KLEE, DART, CUTE) [4, 18, 32] or coverage-guided fuzzing (e.g., AFLplusplus) [15, 37]. While effective at exploring program paths via constraint solving or mutation-based input generation, these approaches have fundamental limitations. For instance, KLEE operates on LLVM bitcode rather than C source code directly, so it requires the code to compile successfully before any analysis can begin. KLEE also requires a manually written driver code that invokes the target function with symbolic inputs. Moreover, it produces raw test input values rather than standalone unit test files with assertions, setup, and cleanup. Similarly, AFLplusplus requires compilable binaries, a manually written fuzz harness, and only produces raw input byte streams rather than structured test code. More broadly, these tools focus on bug detection rather than synthesizing maintainable, self-contained unit tests, and they struggle with path explosion and modeling complex environment interactions. SPARC addresses these limitations by incorporating semantic intent and structured scaffolding through LLM-guided synthesis.

**LLM-Based Unit Test Generation.** Early LLM-based approaches like TestPilot [31] use function signatures and examples to guide zero-shot generation, iteratively repairing failing tests via re-prompting. Subsequent work integrates structured validation-repair loops: ChatUniTest [6] combines context-aware prompting with automated repair to improve correctness and coverage, while TestGen-LLM [2] demonstrates industrial deployment with quality filters and test suite maintenance.

Recent systems incorporate domain knowledge and coverage feedback. KTester [25] uses explicit testing knowledge to guide LLMs toward maintainable tests, and TestART [21] co-evolves generation with dynamic execution signals to reduce hallucination. For Java, Panta [20] iteratively uses control flow and coverage analysis to steer LLMs toward uncovered paths. Empirical studies [1, 7, 29, 38] reveal persistent challenges: undefined path coverage (focus on happy paths), hallucinated dependencies (invented helpers), and lack of traceability.

SPARC distinguishes itself through *explicit intermediate representations*: statement-level CFG-based path enumeration ensures systematic coverage, RAG-constrained operation maps prevent hallucination by retrieving helpers from validated pools, and path-specific synthesis provides direct traceability between tests and execution scenarios.

**LLM-Based Repair and Retrieval-Augmented Generation.** Retrieval-augmented approaches reduce hallucinations by grounding generation in project-relevant context. ReCode [39] retrieves code snippets to improve repair accuracy, while self-debugging methods [5] iteratively refine outputs using internal reasoning or external tool diagnostics. Test-specific frameworks like REAC-CEPT [8], SYNTER [26], and TaRGET [30] repair obsolete tests using compiler errors, test failures, and coverage feedback.

While these systems demonstrate the value of retrieval and iterative validation, they focus on *repairing existing tests* rather than *generating new ones with explicit coverage goals*. SPARC extends this paradigm by using RAG not just for repair, but proactively during synthesis: the operation map retrieves validated helpers *before*

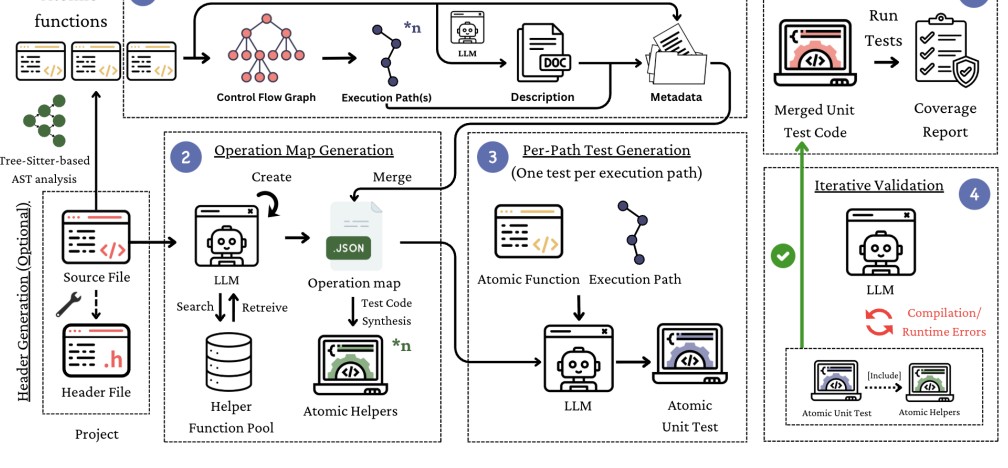

**Figure 3: The four-stage SPARC pipeline. (1) Pre-processing extracts functions, their dependencies, and control-flow paths. (2) RAG-augmented operation map construction retrieves and assigns helper functions per path. (3) Path-targeted synthesis generates a test case for each scenario. (4) Iterative validation compiles, executes, and repairs failing tests.**

generation, constraining the LLM's synthesis space and preventing hallucination at the source.

## 3 Methodology

### 3.1 Architectural Overview

Figure 3 illustrates the architecture of SPARC, a *scenario-based* pipeline that bridges complex C source code and validated unit tests by decomposing test synthesis into per-path scenarios. The pipeline is guided by three principles: *automation* across diverse C projects, *correctness* via iterative validation, and *semantic alignment* through multi-stage reasoning that maps program logic to concrete test scenarios.

### 3.2 Preliminaries and Formal Definitions

To precisely characterize the difference between vanilla prompt generation and SPARC's structured approach, we introduce the following formal definitions.

**Function Under Test.** We define a function under test as a tuple:

$$f = \langle name, \sigma, body, desc, deps \rangle$$

where *name* is the identifier, $\sigma = (\tau_1, \ldots, \tau_n) \to \tau_r$ is the type signature (parameter types and return type $\tau_r$), *body* is the implementation, *desc* is an LLM-generated semantic description, and *deps* is the set of functions called by $f$.

**Control Flow Graph and Execution Paths.** For each function $f$, we construct a Control Flow Graph $\text{CFG}(f) = \langle V, E, v_{\text{entry}}, v_{\text{exit}} \rangle$, where $V$ is the set of basic blocks, $E \subseteq V \times V$ is the set of control flow edges, and $v_{\text{entry}}, v_{\text{exit}} \in V$ are the unique entry and exit nodes. An *execution path* $\pi$ is a sequence of nodes $\pi = (v_{\text{entry}}, v_1, \ldots, v_k, v_{\text{exit}})$ representing one feasible program execution. We denote the set of all extracted paths as $\text{Paths}(f)$.

**Helper Function Pool.** Let $\mathcal{H} = \{h_1, \ldots, h_m\}$ be a curated pool of validated helper functions, where each $h_i = \langle name_i, \sigma_i, impl_i, desc_i \rangle$

encapsulates common testing operations such as data structure initialization, memory cleanup, and Unity assertions.

**RAG-Based Matching.** Each helper's description is embedded into a shared vector space. Given a target function $f$, cosine similarity retrieves the subset of helpers most relevant to its source code:

$$\Lambda(f) = \{h \in \mathcal{H} \mid \text{sim}(\text{embed}(source(f)), \text{embed}(desc(h))) > \theta\}$$

where $\text{sim}(\cdot, \cdot)$ is cosine similarity and $\theta$ is a retrieval threshold. Only the name, description, return type, and parameter signatures of each helper are retrieved and not the implementation bodies.

**Operation Map.** The Operation Map is produced via a single LLM call that receives three inputs: the full source code $source(f)$, the associated header files, and the RAG-retrieved helper catalog $\Lambda(f)$. The LLM outputs:

$$\Omega(f) = \langle \Lambda_{\text{reuse}}, \mathcal{H}_{\text{created}}, deps(f) \rangle$$

where $\Lambda_{\text{reuse}} \subseteq \Lambda(f)$ are existing helpers selected for reuse, $\mathcal{H}_{\text{created}}$ are new helper functions defined by the LLM with full implementations, and $deps(f)$ is a dependency analysis of which source functions $f$ calls. Path information is not provided at this stage. It is merged into the operation map output afterward and consumed during test synthesis.

**Test Suite.** A test suite $\mathcal{T}(f) = \{t_1, \ldots, t_k\}$ is a set of C test functions, each invoking $f$ with inputs that target a specific path and asserting expected outcomes via Unity [23] macros. The goal is full path coverage: $\forall \pi \in \text{Paths}(f), \exists t \in \mathcal{T}(f) : \text{covers}(t, \pi)$.

### 3.3 SPARC Multi-Stage Pipeline

SPARC decomposes test synthesis into four explicit stages (§ 3.3.1– 3.3.4), each producing intermediate artifacts that guide the next. The overall pipeline can be expressed as:

$$\mathcal{T}_{\text{SPARC}}(f) = \text{Merge}\left(\bigcup_{\pi \in \text{Paths}(f)} \text{Validate}\left(\text{LLM}_{\text{SPARC}}(f, \pi, \Omega(f))\right)\right)$$

This formulation highlights the following features of the SPARC:

(1) **Path-based decomposition:** Tests are generated independently for each execution path $\pi \in \text{Paths}(f)$, ensuring systematic coverage.
(2) **Operation Map constraint:** The *operation map* is constructed by a single LLM call over the entire source file.
(3) **Validation loop:** Each generated test undergoes Validate($\cdot$), an iterative compilation and execution process that refines code based on error feedback.

*3.3.1 Pre-processing from Source Code to Structured Metadata.* Pre-processing transforms raw C source code into structured metadata. For each function $f$, this stage computes:

$$\Phi(f) = \langle \text{CFG}(f), \text{Paths}(\text{CFG}(f)), deps(f), desc(f) \rangle$$

**Header Generation and Function Extraction.** Clang [27] pre-processes source files and consolidates function declarations into a project-specific header, ensuring all dependencies $deps(f)$ are properly declared. Tree-sitter [35]-based AST analysis then extracts each target function into an independent source file.

**CFG Construction and Path Extraction.** For each function $f$, the pipeline uses ATLAS [9] to construct its statement-level control flow graph $\text{CFG}(f)$ and enumerate all feasible execution paths $\text{Paths}(\text{CFG}(f))$. Paths extracted from LLVM [24] or GCC [16]-generated CFGs operate on lowered intermediate representations that strip variable names, source-level types, and program structure, rendering them unsuitable for LLMs trained on source code. Moreover, both approaches require compilable code. Since ATLAS natively builds CFGs with inter-functional dependencies spanning the entire project, we modify it to produce isolated statement-level CFGs per function. Each extracted path $\pi \in \text{Paths}(\text{CFG}(f))$ is then linearized into JSON (e.g., START $\rightarrow$ (root==NULL) $\rightarrow$ RETURN), making control-flow intent explicit.

**Semantic Description.** For each function, the pipeline assembles structural metadata $\mathcal{M}(f) = \langle \text{name}(f), \sigma(f), \text{Paths}(f), deps(f) \rangle$ and passes it to an LLM to produce a semantic description $desc(f) = \text{LLM}(\mathcal{M}(f))$ (e.g., "Inserts a value into a BST, maintaining the BST property"). We set LLM temperature to 0 at this stage and throughout all subsequent stages to ensure deterministic outputs. This description guides operation map construction in the next stage.

*3.3.2 Operation Map Construction and Helper Synthesis.* After pre-processing, SPARC jointly produces the Operation Map $\Omega(f)$ and an initial helpers file (helpers.c). First, RAG retrieves the relevant helper subset $\Lambda(f)$ from pool $\mathcal{H}$ by matching the source code of $f$ against helper descriptions via cosine similarity (Section 3.2). A single LLM call then receives $source(f)$, the associated header files, and $\Lambda(f)$, and outputs $\Omega(f)$: which existing helpers to reuse ($\Lambda_{\text{reuse}}$), any new helpers to create ($\mathcal{H}_{\text{created}}$) with full C implementations, and the dependency analysis $deps(f)$. Path information from the pre-processing stage is merged into $\Omega(f)$ afterward.

The helpers file is then assembled: for each helper in $\Lambda_{\text{reuse}}$, the implementation is resolved from the on-disk helper pool; for each helper in $\mathcal{H}_{\text{created}}$, the LLM-generated code is used directly. The resulting helpers.c is duplicated into each per-path atomic unit, yielding an independent copy helpers$_\pi$ for each path $\pi \in \text{Paths}(f)$.

*3.3.3 Per-Path Test Generation.* For each execution path $\pi \in \text{Paths}(f)$, SPARC issues a dedicated LLM call to synthesize a self-contained Unity test file:

$$t_\pi = \text{LLM}_{\text{SPARC}}(f, \pi, \Omega(f)[\pi])$$

The LLM receives the atomic function body $f$, the linearized path $\pi$, and the path-specific operation map $\Omega(f)[\pi]$ as a specification of *all and only* the functions the test may invoke. The generated test constructs input data that triggers $\pi$, calls the function under test, and asserts expected behavior using Unity macros without referencing any function outside the declared operation map.

*3.3.4 Iterative Validation and Test Merging.* Each path $\pi$ produces an atomic unit $(t_\pi, \text{helpers}_\pi)$ that is validated jointly. Let

$$\text{Errors}(t_\pi, \text{helpers}_\pi) = \text{compile}(t_\pi, \text{helpers}_\pi) \cup \text{run}(t_\pi, \text{helpers}_\pi)$$

where compile($\cdot$) returns static errors (undefined references, type mismatches) and run($\cdot$) returns dynamic errors detected by AddressSanitizer [33] (memory leaks, buffer overflows, segmentation faults). The LLM applies targeted fixes to the pair:

$$(t'_\pi, \text{helpers}'_\pi) = \text{Fix}(t_\pi, \text{helpers}_\pi, \text{Errors}(t_\pi, \text{helpers}_\pi)).$$

The LLM temperature is raised to 0.1 during iterative repair to allow slight variation across retry attempts. Because each atomic unit test is repaired independently, the validated helpers$'_\pi$ may differ across paths. Unit tests that fail to stabilize after $N = 3$ iterations are discarded.

Validated unit tests are then merged into a unified suite:

$$\mathcal{T}_{\text{SPARC}}(f) = \text{Merge}(\mathcal{T}_{\text{validated}}(f)),$$

$$\mathcal{T}_{\text{validated}}(f) = \{t_\pi \mid \pi \in \text{Paths}(f), \text{Errors}(t_\pi, \text{helpers}'_\pi) = \emptyset\}.$$

The Merge($\cdot$) operator deduplicates headers and helper functions across all validated units, then concatenates the test bodies. The final suite is compiled with gcov instrumentation to measure line and branch coverage.

# 4 Evaluation

We assess SPARC to answer the following six research questions:

- **RQ1: Coverage Effectiveness.** (RQ1.1) How does SPARC compare to existing test generation approaches (symbolic execution, vanilla prompting) on code coverage? (RQ1.2) How does coverage vary across subjects of different complexity?
- **RQ2: Test Validity and Fault Detection.** (RQ2.1) What proportion of generated tests are retained after iterative validation and repair? (RQ2.2) How effective are the retained tests at detecting faults, as measured by path diversity and mutation score?
- **RQ3: Root Causes of Failures.** (RQ3.1) What are the categories of errors in tests that fail to compile or execute correctly? (RQ3.2) How effective is iterative LLM-based repair at resolving each failure category?
- **RQ4: Perceived Test Quality.** How do developers perceive the quality of SPARC-generated tests compared to baseline LLM-generated tests?
- **RQ5: Scalability and Cost Analysis.** (RQ5.1) How does SPARC's generation cost (API tokens) scale with subject complexity? (RQ5.2) What is the per-test cost breakdown?
- **RQ6: LLM Choice Impact.** How does the choice of underlying LLM affect the effectiveness of SPARC's test generation?

**Table 1: Comparing SPARC with DeepSeek V3.2 (DS), and KLEE (subjects arranged in descending order of LOC; only the top 15 bigger subjects are shown independently and the combined results of all subjects are in the last row).**

| Dataset | Subjects | #Control-flow Paths | #Function | #Line | #Branch | Function Cov. (%) | | | Line Cov. (%) | | | Branch Cov. (%) | | | Mutation Score (%) | |
|---|---|---|---|---|---|---|---|---|---|---|---|---|---|---|---|---|
| | | | | | | SPARC | DS | KLEE | SPARC | DS | KLEE | SPARC | DS | KLEE | SPARC | DS |
| Rustine | heman | 493 | 298 | 8,385 | 1,581 | 65.10 | 53.02 | 61.41 | 47.27 | 15.69 | 33.77 | 20.18 | 10.62 | 18.22 | 39 | 9 |
| | lodepng | 2,420 | 225 | 6,328 | 2,853 | 65.33 | 40.00 | 64.89 | 42.00 | 17.04 | 37.60 | 20.01 | 5.68 | 16.86 | 28 | 15 |
| | tulipindicators | 517 | 269 | 4,706 | 2,446 | 83.64 | 53.16 | 63.36 | 59.57 | 21.31 | 45.23 | 25.27 | 5.52 | 18.40 | 13 | 3 |
| | genann | 26 | 15 | 176 | 110 | 100.00 | 100.00 | 100.00 | 92.60 | 80.10 | 94.20 | 80.90 | 65.50 | 68.80 | 64 | 42 |
| | rgba | 45 | 13 | 95 | 126 | 100.00 | 92.30 | 100.00 | 96.80 | 81.10 | 93.00 | 75.40 | 63.50 | 68.10 | 83 | 30 |
| TheAlgorithms C | red_black_tree | 55 | 11 | 309 | 206 | 100.00 | 70.00 | 100.00 | 90.00 | 21.70 | 82.00 | 91.30 | 15.00 | 43.80 | 67 | 15 |
| | multikey_quick_sort | 74 | 27 | 301 | 196 | 100.00 | 41.20 | 100.00 | 86.40 | 34.50 | 77.60 | 92.90 | 39.00 | 65.80 | 39 | 20 |
| | mcnaughton_yamada_thompson | 16 | 22 | 261 | 128 | 95.50 | 88.00 | 100.00 | 84.70 | 87.10 | 92.00 | 74.20 | 88.60 | 97.00 | 58 | 66 |
| | tic_tac_toe | 20 | 7 | 164 | 206 | 87.10 | 65.60 | 100.00 | 85.10 | 55.00 | 96.40 | 85.10 | 57.20 | 94.00 | 76 | 37 |
| | naval_battle | 91 | 6 | 156 | 224 | 100.00 | 83.30 | 100.00 | 93.60 | 55.50 | 93.60 | 82.10 | 52.30 | 76.00 | 39 | 32 |
| | non_preemptive_priority | 33 | 11 | 148 | 72 | 81.80 | 81.80 | 100.00 | 86.50 | 83.10 | 94.80 | 81.90 | 75.00 | 91.40 | 83 | 75 |
| | avl_tree | 76 | 16 | 131 | 78 | 100.00 | 94.10 | 100.00 | 96.20 | 60.50 | 91.50 | 89.70 | 68.40 | 81.40 | 59 | 55 |
| | hash_blake2b | 26 | 6 | 94 | 38 | 100.00 | 75.00 | 100.00 | 98.90 | 70.90 | 96.80 | 97.40 | 72.90 | 70.30 | 24 | 13 |
| | adaline_learning | 18 | 7 | 53 | 28 | 100.00 | 10.00 | 100.00 | 92.50 | 0.60 | 93.10 | 78.60 | 3.10 | 45.80 | 39 | 7 |
| | kohonen_som_trace | 14 | 5 | 49 | 32 | 100.00 | 16.70 | 100.00 | 100.00 | 12.30 | 96.60 | 100.00 | 15.00 | 60.60 | 63 | 4 |
| Total/Average | All 59 Subjects | 5,233 | 1,461 | 26,302 | 11,855 | 94.88 | 68.94 | 98.13 | 89.36 | 58.00 | 82.63 | 81.85 | 55.84 | 64.13 | 58.25 | 37.47 |

## 4.1 RQ1: Coverage Effectiveness

*4.1.1 Experimental Setup.* We evaluate SPARC using algorithmic subjects from TheAlgorithms C repository [34]. Specifically, we select the 51 largest C projects, measured by lines of code. We also select eight subjects from Rustine [13], which collects real-world C projects from several C-to-Rust translation works. To maintain consistency, we exclude the default main and test harness functions from each source program, as these serve as scaffolding rather than containing the core algorithmic function under test. Additionally, we remove all I/O operation functions, functions requiring any mocking, and convert static functions to non-static to make them accessible from external test files.

**Baselines.** We compare SPARC (DeepSeek V3.2 as the underlying LLM) against two baselines[2]: (1) KLEE [4], a widely used symbolic execution engine for C. (2) DeepSeek V3.2 (DS), vanilla LLM prompting that receives the source code (the function under test and its surrounding class as the context) in a single prompt and is instructed to generate tests for all code paths and edge cases targeting 100% code coverage, but without CFG-guided path enumeration, operation map, or iterative validation.

**Evaluation Metrics.** We adopt commonly used quality metrics in test generation literature, including function, line, and branch coverages. Since we evaluate at the project level rather than the method level, we measure the metrics per project and report the average coverage across all target projects.

*4.1.2 Coverage Comparison Against Existing Approaches.* Based on industry standards [3, 17], typical code coverage targets range from 70% to 80%. Google, for instance, classifies 75% as "commendable," [22]. For brevity, Table 1 shows 15 representative subjects in detail. The summary row averages over all 59 benchmark subjects. In terms of line and branch coverage, SPARC consistently outperforms DS in almost all projects. On average, SPARC achieves 31.36% better line and 26.01% better branch coverage than DS. Moreover, the performance is on-par with KLEE in most projects and even exceeds it in the larger projects (heman, red_black_tree etc.).

> **Finding:** SPARC outperforms DS on average in both line and branch coverage, exceeding KLEE on the complex subjects.

*4.1.3 Coverage vs. Subject Complexity.* To understand how coverage varies with subject complexity, we use the number of control-flow paths as a complexity measure and partition the subjects into three tiers: *low* (≤10 paths; 3 subjects), *medium* (11–30 paths; 23 subjects), and *high* (>30 paths; 33 subjects).

**The advantage over DS widens with complexity.** For DS, the mean branch coverage falls from 86% in the low tier to ≈67% in the medium and ≈45% in the high tier. The branch coverage gap between SPARC and DS consequently widens from 14% for low to approximately 20% and 30% for medium and high-complexity subjects respectively. Without explicit path guidance, DS struggles to reach structurally deeper branches, causing its coverage to plateau on complex subjects while SPARC continues to exercise new logic.

**SPARC is resilient to increasing complexity.** In the low tier, SPARC achieves perfect line and branch coverage on all three subjects. Coverage remains high in the medium tier (94.4% line, 86.9% branch) and in the high tier (88.2% line, 76.3% branch) despite subjects containing up to 2,420 paths. Across all subjects, the Spearman rank correlation between path count and SPARC branch coverage is moderate but not statistically significant ($\rho=-0.39$, $p=0.12$) to conclusively indicate a negative trend.

> **Finding:** SPARC's coverage shows a statistically insignificant correlation with path count, while the gap over DS widens with increased complexity.

## 4.2 RQ2: Test Validity and Fault Detection

*4.2.1 Test Retention Through Iterative Repair.* To evaluate the correctness of SPARC-generated tests and the effectiveness of the iterative validation loop, we track each test from generation through repair on a subset of 10 projects[3]. For each project, Table 2 shows the number of generated tests, tests that pass without any repair (Pass$_0$), tests that enter the repair loop (Fail), tests that are fixed per iteration, and tests it ultimately drops or keeps in the final suite.

**Test correctness.** Of 282 generated tests, 235 (83.3%) compile and pass on the first attempt. The remaining 47 enter the iterative repair loop: 10 fail due to compilation errors and 37 due to runtime issues.

---

[2]To the best of our knowledge, existing neuro-symbolic test generation pipelines are either for Java and Python, or they are not publicly available.

[3]To manually analyze and identify the exact issues resulting in test failure, we select smaller projects (LoC) where SPARC achieves high line and branch coverage (RQ1). The same subjects are also used in RQ6, since DS initially generates tests with high coverage, and switching to a distilled LLM can track the exact performance loss.

**Table 2: Test validation results per subject.**

| Subjects | #Tests Generated | #Tests $Pass_0$ | #Tests Failed | Repair Iterations 1 | 2 | 3 | #Tests Dropped | #Tests Final |
|---|---|---|---|---|---|---|---|---|
| alaw | 17 | 15 | 2 | 1 | 1 | 0 | 0 | 17 |
| affine | 25 | 20 | 5 | 4 | 1 | 0 | 0 | 25 |
| decimal_to_any_base | 21 | 20 | 1 | 1 | 0 | 0 | 0 | 21 |
| infix_to_postfix | 35 | 32 | 3 | 3 | 0 | 0 | 0 | 35 |
| lcs | 20 | 20 | 0 | 0 | 0 | 0 | 0 | 20 |
| ascending_priority_queue | 36 | 32 | 4 | 3 | 0 | 0 | 1 | 35 |
| bst | 26 | 20 | 6 | 3 | 3 | 0 | 0 | 26 |
| doubly_linked_list | 12 | 4 | 8 | 3 | 1 | 1 | 3 | 9 |
| dynamic_stack | 61 | 46 | 15 | 4 | 0 | 1 | 10 | 51 |
| prime_factorization | 29 | 26 | 3 | 0 | 1 | 0 | 2 | 27 |
| **Total** | **282** | **235** | **47** | **22** | **7** | **2** | **16** | **266** |

Of the 37 runtime failures, *Memory* errors account for 13, *Crash*es for 7, and *Assertion* mismatches for 1. The remaining 16 stem from miscellaneous runtime faults. The loop fixes 31 of these 47, yielding a final suite of 266 tests (94.3% retention).

> **Finding:** SPARC retains 94.3% of generated tests after repair.

*4.2.2 Fault Detection Effectiveness.* We also check whether each retained test adds unique value. 75% of SPARC's tests follow at least one unique control-flow path, compared to 67% for DS. This shows that CFG-guided generation avoids re-testing already-covered logic, which in turn helps kill more distinct mutants.

**Mutation score.** We use Mull [14] to calculate mutation scores as shown in Table 1. SPARC demonstrates 20.78% improvement in average mutation scores over DS. The largest gains appear on subjects where DS has low coverage, e.g., red_black_tree (67% vs. 15%), since unreached code cannot kill mutants. However, coverage alone does not explain the full gap. On qsort, both methods reach 100% line and branch coverages, yet SPARC still shows a 6% higher mutation score (87% vs. 81%), suggesting its scenario-guided generation also produces stronger test oracles.

> **Finding:** SPARC achieves a higher average mutation score than DS, driven by structural coverage. Even at equal coverage, it produces stronger test oracles.

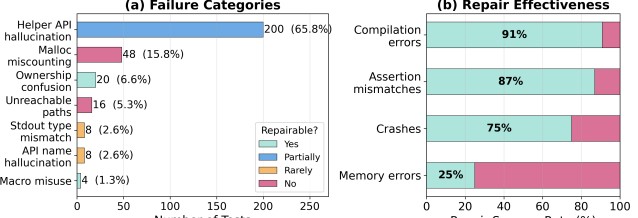

**Figure 4: Root-cause analysis of 304 permanently dropped tests. (a) Failure categories. (b) Repair success rate.**

## 4.3 RQ3: Root Causes of Failures

*4.3.1 Failure Categories.* To understand *why* some tests fail at scale, we extend the analysis to the full benchmark of 59 programs and manually classify the 304 tests that are permanently dropped due to LLM-related errors even after three repair iterations. We inspect compiler diagnostics, sanitizer output, and the generated test code against the corresponding source programs, yielding seven root-cause categories. Figure 4(a) shows the distribution of these categories, ordered by prevalence and color-coded by repairability.

**Helper API hallucination is the dominant failure mode.** As Figure 4(a) shows, helper API hallucination accounts for nearly two-thirds of all dropped tests. The LLM generates tests calling SPARC-generated helper functions with incorrect signatures (e.g., passing three arguments to a two-parameter function in genann), hallucinates function names (e.g., get_size() instead of the actual len() in vector), or references functions that are planned in the operation map but are never generated.

**Memory-related reasoning errors are the second-largest category.** Malloc counter miscounting (15.8%) occurs when the LLM confuses cumulative allocations with allocations occurring *after* a malloc_fail_after() call, causing fault-injection tests to set incorrect failure points. For instance, tests for dynamic_stack's copy_stack() set malloc_fail_after(2) intending to "skip past" allocations that already occur before the call, but the counter resets to zero upon the function call, allowing two more allocations to succeed and the test assertion to fail. Memory ownership confusion (6.6%) occurs when the test frees memory that the function under test has already freed, or vice versa, causing AddressSanitizer to flag the invalid access.

**Remaining categories.** Unreachable path conditions (5.3%) occur when the path extractor selects dead code that can never execute, such as a final else return 0 in a function that already covers all cases with >, <, and ==. Source API name hallucination (2.6%) occurs when the LLM renames valid C identifiers to avoid C++ keyword conflicts (e.g., delete → delete_node). Stdout capture type mismatches (2.6%) occur when the helper API returns an int * pointer but the LLM writes test code expecting the actual data structure (e.g., struct node *), causing a type error. Unity macro misuse (1.3%) involves using the wrong assertion macro (e.g., TEST_ASSERT_EQUAL for pointers, which truncates 64-bit addresses to 32 bits).

> **Finding:** The seven root-cause categories, in order of prevalence, are: helper API hallucination, malloc counter miscounting, memory ownership confusion, unreachable paths, source API name hallucination, stdout type mismatches, and Unity macro misuse.

*4.3.2 Iterative Repair Effectiveness.* We analyze repair effectiveness along two dimensions: per-failure-type repair rates and cross-category iteration efficiency across the full benchmark. Figure 4(b) summarizes repair success rates by failure type.

**Repair rates vary sharply by failure type.** As Figure 4(b) shows, compilation errors and assertion mismatches can be repaired at high rates (91% and 87% respectively), and crashes at 75%, but memory errors at only 25%. This gap exists because compilation and assertion errors give clear error messages that guide the repair agent, while memory errors only describe symptoms like "detected memory leaks" without pointing to the underlying cause. No malloc counter miscounting test is repaired, since the repair agent fixes cleanup code instead of correcting the allocation count itself. However, a high repair rate does not always mean high-quality tests. For example, in tulipindicators, the LLM defaults to trivial inputs (e.g., empty arrays or invalid options) that only trigger early-return checks like if (size <= start) return TI_OKAY. The repair agent fixes the assertion to match this trivial output, so the test passes but never exercises the actual algorithm.

**Repair converges rapidly when the failure is repairable.** Among repaired tests, 77.4% are fixed in the first attempt, 19.4%

Table 3: Participant profile distribution.

| Software Engineer (Experience) | | | Graduate Student | Total Participants |
|---|---|---|---|---|
| <1 yr | 1–3 yrs | 3–5 yrs | | |
| 2 | 2 | 1 | 5 | 10 |

in the second, and only 3.2% in the third, indicating that repairs converge quickly when the failure is repairable.

**Repair cost varies significantly across subjects.** In total, `max_heap` uses 16 iterations on recurring helper dependency errors with little success. In contrast, genann requires a total of 39 iterations to fix 24 tests, because the agent fixes only one bug per iteration while facing multiple independent issues.

> **Finding:** Iterative repair can fix *how* a test fails but cannot fix *what* a test covers. Repairs converge quickly, when error messages are clear.

### 4.4 RQ4: Perceived Test Quality

*4.4.1 Motivation.* Since automated metrics alone do not capture the practical quality of test code, we conduct a user study to assess whether SPARC-generated tests better align with developer expectations than vanilla DS-generated tests.

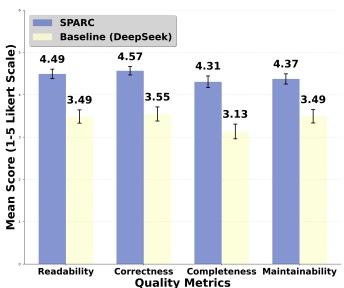

**Figure 5: Perceived unit test survey results: SPARC vs. vanilla DS (95% confidence intervals).**

*4.4.2 Study Procedure and Evaluation Criteria.* We conduct a blind A/B user study in which participants (details in Table 3) compare C unit tests generated by SPARC and DS. For each task, participants inspect a target C function alongside two anonymized test snippets in randomized order and rate each independently along four quality dimensions using a 5-point Likert scale. The questionnaire defines the evaluation criteria as follows: **Readability** measures how easily a developer can follow the test intent and logic, rated from 1 (Very hard to follow) to 5 (Very intuitive). **Correctness** measures whether the test accurately verifies the intended behavior of the target function, rated from 1 (Full of errors) to 5 (Totally correct). **Completeness** measures whether the tests cover edge cases and boundary conditions beyond the happy path, rated from 1 (Only happy path) to 5 (Comprehensive coverage). **Maintainability** measures how easy it is to extend the test code as requirements change, rated from 1 (Very difficult) to 5 (Very easy).

*4.4.3 Results.* Figure 5 summarizes mean ratings across all four dimensions based on 150 paired comparisons per dimension (10 participants evaluating between 10 and 20 tasks each). Paired $t$-tests show SPARC significantly outperforms the baseline in Readability ($\Delta = 1.01$, $t(149) = 9.86$, $p<0.001$, $d = 0.81$), Correctness ($\Delta = 1.02$, $t(149) = 9.99$, $p<0.001$, $d = 0.82$), Completeness ($\Delta = 1.17$, $t(149) = 10.61$, $p<0.001$, $d = 0.87$), and Maintainability ($\Delta = 0.88$, $t(149) = 8.57$, $p<0.001$, $d = 0.70$). Effect sizes are large for the

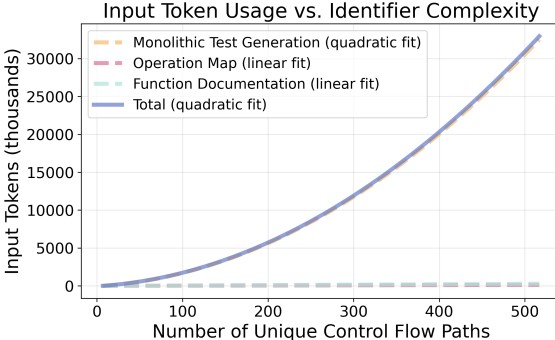

**Figure 6: Token usage vs. path count.**

first three dimensions and medium for Maintainability, and the improvements are consistent across all tasks and dimensions.

*4.4.4 Reliability and Validity Considerations.* Inter-rater reliability is assessed using Krippendorff's Alpha computed on per-task preference scores, yielding $\alpha = 0.32$ for Readability and $\alpha$ values between 0.08 and 0.23 for the remaining dimensions. These values indicate limited absolute agreement, which is expected for two reasons. First, participants apply different personal standards when judging code quality, and some participants evaluate fewer tasks (between 10 and 20), further reducing cross-rater overlap. Second, the two approaches generate tests targeting different execution paths for the same function, so raters compare structurally different code. Despite these sources of variation, the paired design mitigates scale bias by comparing both approaches on the same function under test and rater, and the paired $t$-tests yield a robust and consistent signal of relative preference across all four dimensions.

### 4.5 RQ5: Scalability and Cost Analysis

*4.5.1 Cost Scaling.* To understand how SPARC's token consumption scales with project complexity, we measure the input tokens consumed by each pipeline stage against the number of unique control-flow paths across all subjects. We focus on *forward-pass* tokens shown in Figure 6; those consumed by the Operation Map, Function Documentation, and Per-Path Test Generation (hereafter *Monolithic Test Generation*) and analyze the Validation stage separately, as its cost depends on the number of failed tests. The three forward-pass stages exhibit different scaling behavior.

**Constant-Cost Stages.** The Operation Map and Function Documentation stages together account for only 8.7% of the total forward-pass cost on average. The Operation Map stage requires exactly one API call per subject (mean 6,664 tokens), while Function Documentation scales with the number of functions, averaging 11,044 tokens per subject.

**Dominance of Monolithic Test Generation.** The Monolithic Test Generation stage dominates cost, accounting for 91.3% of the forward-pass total on average. To avoid redundant coverage, this stage issues one LLM call per control-flow path. This accumulation causes the per-path cost to grow as the test suite expands. As shown in Figure 6, excluding lodepng, the monolithic stage's total cost is well-approximated by a quadratic fit ($r = 0.999$, $R^2 = 0.998$), and the rising per-path overhead means the total forward-pass cost is also captured better by a quadratic fit ($R^2 = 0.998$).

**Table 4: Coverage comparison across `DeepSeek V3.2`, `Gemini 3 Flash Preview`, and `GPT-5-Mini`.**

| Subject | #Paths | #Function | #Line | #Branch | Function Cov. (%) | | | Line Cov. (%) | | | Branch Cov. (%) | | | Mutation Score (%) | | |
|---|---|---|---|---|---|---|---|---|---|---|---|---|---|---|---|---|
| | | | | | DeepSeek | Gemini 3 Flash | GPT-5-Mini | DeepSeek | Gemini 3 Flash | GPT-5-Mini | DeepSeek | Gemini 3 Flash | GPT-5-Mini | DeepSeek | Gemini 3 Flash | GPT-5-Mini |
| alaw | 17 | 11 | 74 | 34 | **100.00** | **100.00** | **100.00** | 95.90 | **100.00** | **100.00** | 82.40 | **100.00** | 97.06 | 62 | **77** | 65 |
| affine | 25 | 12 | 95 | 40 | **100.00** | **100.00** | **100.00** | 86.30 | 96.84 | **100.00** | 85.00 | **95.00** | **95.00** | 44 | 59 | **74** |
| decimal_to_any_base | 21 | 9 | 61 | 40 | **100.00** | **100.00** | **100.00** | **95.10** | 95.08 | 95.08 | **72.50** | 70.00 | 70.00 | 72 | **75** | **75** |
| infix_to_postfix | 35 | 14 | 117 | 110 | **100.00** | **100.00** | **100.00** | 79.50 | 79.49 | **80.34** | 69.10 | **70.00** | **70.00** | **76** | 65 | 70 |
| lcs | 20 | 11 | 100 | 76 | **100.00** | **100.00** | **100.00** | 93.00 | **95.00** | 91.00 | **85.50** | 75.00 | 71.05 | **69** | 58 | 57 |
| ascending_priority_queue | 36 | 15 | 143 | 92 | **100.00** | 86.67 | **100.00** | 91.60 | 72.03 | **93.01** | 79.30 | 61.96 | **79.35** | 69 | **74** | **74** |
| bst | 26 | 8 | 76 | 44 | **100.00** | **100.00** | **100.00** | **98.70** | 98.68 | 98.68 | **93.20** | 93.18 | 90.91 | **74** | **74** | 69 |
| doubly_linked_list | 12 | 5 | 79 | 44 | **100.00** | **100.00** | **100.00** | **97.40** | 77.22 | 58.23 | **86.40** | 65.91 | 43.18 | **81** | 56 | 37 |
| dynamic_stack | 61 | 16 | 161 | 128 | **93.80** | 87.50 | 87.50 | 85.10 | **87.58** | 86.96 | 68.80 | **70.31** | 69.53 | **70** | **70** | 69 |
| prime_factorization | 29 | 9 | 126 | 76 | **100.00** | **100.00** | **100.00** | **76.20** | 70.63 | 73.81 | 75.00 | **76.32** | 75.00 | 52 | 55 | **57** |
| **Total/Average** | 284 | 110 | 1,032 | 684 | **99.38** | 97.42 | 98.75 | **89.88** | 87.26 | 87.71 | **79.72** | 77.77 | 76.11 | **66.90** | 66.30 | 64.70 |

**Variance at High Complexity.** Considerable variance emerges for subjects with more than 50 paths. For example, `avl_tree` (76 paths) and `multikey_quick_sort` (74 paths) consume more than 1M total tokens each, while `min_heap` (63 paths) consumes only 885K tokens. The two expensive subjects use macros, shared variables, and deeply nested control paths, all of which inflate prompt size. In contrast, `min_heap` uses a flat array representation with simple branching and linear control flow. Thus, path count alone is an incomplete predictor of token cost. The structural complexity per path is an equally important factor.

> **Finding:** SPARC's token cost grows roughly quadratically with the number of control-flow paths, but path count alone is an incomplete predictor. The structural complexity per path is an equally important cost driver.

*4.5.2 Cost Breakdown Per-Test.* We normalize token consumption by the number of successfully generated tests per subject, this time including the Validation stage as well. Across all subjects, SPARC produces tests consuming a mean of 12,152 tokens per test. The interquartile range spans 6,481–15,476 tokens per test, though tests in outliers such as `avl_tree` and `red_black_tree` with structurally complex code reach up to 52,330 tokens.

Input token count per-test is distributed as follows: Operation Map and Function Documentation (6.3%), Monolithic Test Generation (71.6%) and Validation (22.1%). The correlation between test count and per-test cost is weak ($r = 0.15$), which confirms that cost is governed by project complexity rather than suite size.

> **Finding:** SPARC's per-test token cost varies with project structural complexity rather than merely test-suite size.

## 4.6 RQ6: LLM Choice Impact

Since token cost is a practical concern, we assess whether the pipeline maintains its effectiveness with cost-efficient models. We re-run SPARC on the 10-project subset from RQ2 using two distilled frontier LLMs: `Gemini 3 Flash Preview` [19] (temperature = 0, thinking_budget = 0, max_output_tokens = 8192) and `GPT-5-Mini` (gpt-5-mini-2025-08-07) [28] (temperature = 1, max_completion_tokens = 8192), and compare the resulting coverage and mutation scores against `DeepSeek V3.2`. We also evaluate a locally hosted `Qwen3 8B` via Ollama, but it fails to produce valid structured outputs, indicating the pipeline requires a minimum model capability threshold. Table 4 reports per-project results for the three successful models across all four metrics.

**Aggregate performance is tightly clustered.** Across the ten projects, all three models produce comparable averages on every metric. `DeepSeek V3.2` leads on all four even though the margins are narrow. The maximum spread between the best and worst model on any single aggregate metric is only 3.61% (branch coverage).

**Subject differences reflect test-generation choices.** Although the aggregate differences are small, individual subjects can exhibit larger spreads. On `doubly_linked_list`, `DeepSeek V3.2` produces dedicated tests that exercise the nested pointer-relinking branches in functions `insert()` and `delete()`; `GPT-5-Mini` tests only boundary cases leaving all interior branches uncovered. On `ascending_priority_queue` (Gemini 61.96% vs. ≈79% for the other two), all of `Gemini`'s `removes()` tests insert values in ascending order, so the minimum is always the front node; the rear-node and middle-node deletion branches are never exercised. These divergences are subject-specific rather than model-specific. No model consistently underperforms, and the direction of the advantage shifts across subjects.

**Implications for practical deployment.** Both distilled models match `DeepSeek V3.2`'s coverage and mutation scores demonstrating that SPARC's structured pipeline, including statement-level CFG-guided path enumeration, scenario-based prompting, and iterative validation, closes the capability gap between cost-efficient and frontier models. Practitioners can therefore select the underlying model based on cost, latency, or API availability without sacrificing coverage or fault-detection effectiveness, making SPARC viable for resource-constrained environments.

> **Finding:** SPARC's pipeline architecture, not the underlying LLM, is the dominant factor in test quality.

## 5 Conclusion

We present SPARC, a scenario-based framework for automated C unit test generation. SPARC addresses key limitations of vanilla prompting, including low branch coverage, hallucinated dependencies, and lack of traceability, by decomposing each function into per-path scenarios guided by operation maps and iterative repair. Our evaluation on 59 real-world C projects shows that SPARC achieves 31.36% higher line coverage and 26.01% higher branch coverage than vanilla prompting, with a 20.78% improvement in mutation scores, matching or exceeding KLEE on complex subjects. Iterative validation retains 94.3% of generated tests, and a developer study confirms higher ratings in readability, correctness, completeness, and maintainability. Notably, cost-efficient models achieve comparable results to frontier models, indicating that the structured pipeline, not the underlying model, drives test quality. These results show that scenario-guided test generation is a practical approach for automated unit testing of complex C software.

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
