# OpenReview forum: "SPARC: Scenario Planning and Reasoning for Automated C Unit Test Generation"
_ACM.org/AIWare/2026/Conference — Submitted to AIware 2026_

### Official Review · Reviewer_hFR5 · 2026-03-03

**Rating:** 2
**Confidence:** 4

**Review:**

Strengths:
1. The paper is well-written and clearly structured, easy to follow.
2. The research is relatively thorough. The authors evaluate on 59 real-world and algorithmic C subjects and demonstrate that SPARC achieves higher line coverage, branch coverage, and mutation score than the baseline, matching or exceeding KLEE on larger subjects.
3. The four-stage pipeline design is reasonable, and the idea of decomposing test generation into per-path scenarios with operation map grounding is a meaningful contribution.

Weakness:
1. The vanilla prompting baseline does not include iterative repair, while SPARC does. This means the reported improvements (31.36% line coverage, 26.01% branch coverage) conflate the contribution of scenario planning with the contribution of the repair loop. An ablation baseline of "vanilla DS + iterative repair (without CFG guidance)" is necessary to isolate the actual benefit of CFG-guided scenario planning. Without this ablation, the core claim is not adequately supported.
2. The paper does not compare against any existing LLM-based test generation systems (e.g., ChatUniTest, ASTER). The authors should discuss or experimentally demonstrate whether SPARC's scenario planning still provides significant incremental gains if ChatUniTest's "context extraction + iterative repair" logic were adapted to C as another baseline.
3. The abstract claims "a scalable path for industrial-grade testing of legacy C codebases", but the evaluation subjects are small-scale (the largest, lodepng, has only ~8K lines), and the TheAlgorithms/C subjects are educational algorithmic code. The experimental setup also explicitly excludes main functions, I/O functions, functions requiring mocking, and converts static functions to non-static. These exclusions limit the generalizability of the results to real industrial C codebases.
4. The user study has statistical concerns. The sample size is small (n=10), and Krippendorff's Alpha ranges from 0.08 to 0.32, indicating poor inter-rater agreement. Furthermore, Likert-scale data is ordinal, so paired t-tests may not be appropriate — non-parametric tests such as Wilcoxon signed-rank would be more suitable.
5. Although the framework uses statement-level CFG path enumeration, the number of execution paths can grow exponentially in large-scale functions with many loops and complex conditional branches. The paper does not sufficiently discuss how to handle very large functions.
6. The framework depends on static analysis tools like ATLAS for CFG construction. If the source code contains complex macro definitions or platform-specific extensions, the accuracy of static analysis will directly affect the quality of subsequent LLM reasoning.

Questions:
1. The iterative repair stage raises LLM temperature from 0 to 0.1. What is the basis for choosing this value? Has an ablation study been conducted on different temperature values and their effect on repair success rate?
2. KLEE requires manually written driver code, while SPARC is fully automated. How was the KLEE driver code written in the experiments? Was its quality controlled to avoid underestimating KLEE's performance due to poor driver code?
3. The experiments show SPARC outperforms KLEE on complex subjects. Can you provide case studies that analyze which types of subjects benefit from scenario-based LLM reasoning versus traditional constraint-solving symbolic execution?

**Summary:**

This paper proposes SPARC, a neuro-symbolic framework that addresses the leap-to-code failure mode in LLM-based automated C unit test generation. The framework decomposes test generation into a scenario-based reasoning process through four stages: CFG analysis, operation map construction, path-targeted synthesis, and iterative repair. It is evaluated on 59 real-world and algorithmic C subjects.

---

> ### Author Response · Authors · 2026-03-19
>
> **Addressing the weaknesses:**
>
> We thank the reviewer for the constructive feedback. We address your specific concerns below:
>
> **W1 Response:**
>
> $\underline{\text{The DeepSeek baseline does include iterative repair as errors are fed back to DS up to 16 times per project to repair the failed tests}}$. This is done with a generic prompt. The confusing phrase "without iterative validation" in Section 4.1.1 refers to SPARC's specific loop architecture, not the absence of repair. Moreover, Section 4.1.3 demonstrates the pattern of increasing code coverage gap as the project's complexity grows, which stems from CFG-guided path decomposition, not the repair loop.
>
> **W2 Response:**
>
> ChatUniTest and ASTER target Java and Python, both garbage-collected languages where memory is managed automatically. $\underline{\text{A}}$ $\underline{\text{core}}$ $\underline{\text{challenge}}$ $\underline{\text{of}}$ $\underline{\text{C}}$ $\underline{\text{unit}}$ $\underline{\text{test}}$ $\underline{\text{generation,}}$ $\underline{\text{and}}$ $\underline{\text{a}}$ $\underline{\text{central}}$ $\underline{\text{focus}}$ $\underline{\text{of}}$ $\underline{\text{SPARC,}}$ $\underline{\text{is}}$ $\underline{\text{reasoning}}$ $\underline{\text{about}}$ $\underline{\text{manual}}$ $\underline{\text{memory}}$ $\underline{\text{management:}}$ $\underline{\text{malloc/free}}$ $\underline{\text{pairing,}}$ $\underline{\text{pointer}}$ $\underline{\text{ownership,}}$ $\underline{\text{and}}$ $\underline{\text{memory}}$ $\underline{\text{lifecycle}}$ $\underline{\text{as}}$ $\underline{\text{shown}}$ $\underline{\text{in}}$ $\underline{\text{RQ3}}$, Figure 4 analysis. These failure modes are C-specific making a direct adaptation of ChatUniTest or ASTER to C non-trivial. We would also like to mention that the ASTER is not publicly available.
>
> **W3 Response:**
>
> $\underline{\text{We want to clarify that SPARC operates at a per-function level instead of the full project (although it does account for intra- and inter- procedural dependencies between functions in the execution path)}}$. The pre-processing stage extracts each target function into an independent source file (Section 3.3.1, Figure 3). Regarding subject selection, the exclusions reflect standard methodology, not limitations of the approach. Main/test-harness functions are scaffolding, not algorithmic logic. I/O functions require environment mocking orthogonal to unit test generation. The static functions are converted to non-static solely to make them accessible from external test files and increase testable coverage. These are the same exclusions applied in prior C testing work using KLEE. Finally, while we acknowledge that the subjects in our benchmark are mostly small-to-medium in size, lodepng contains 2,420 control-flow paths where KLEE's constraint solver begins to struggle compared to SPARC (Table 1).
>
> **W4 Response:**
>
> Sample size: The statistical unit is the paired comparison, not the participant. Each of the 10 participants evaluated 10-20 function-level tasks, yielding 150 paired comparisons per dimension. Effect sizes range from medium (d = 0.70) to large (d = 0.87) with p < 0.001 across all dimensions in Section 4.4.3.
>
> Krippendorff's Alpha: The low Krippendorff's Alpha reflects differences in how individual raters use the five-point scale rather than a genuine disagreement over which approach is better. Our paired design accounts for this because each rater evaluates both approaches for the same task, any personal bias in their scoring style cancels out when we look at the relative difference between the two.
>
> $\underline{\text{Statistical test choice: Paired t-tests on Likert data are standard in SE user studies. Given the effect sizes, we expect the conclusions to hold in Wilcoxon signed-rank tests.}}$
>
> **W5 Response:**
>
> $\underline{\text{As described in Section 3.3.1, SPARC extracts each target function into an independent source file and generates tests independently per unique control-flow path (Section 3.3.3)}}$, so path explosion within any single function does not propagate across the project. Hence, scaling is linear in the number of functions.

---

> ### Author Response · Authors · 2026-03-19
>
> **Addressing the questions:**
>
> **Q1 Response:**
>
> We initially tried temperature 0 for iterative repair but if the first repair fails, subsequent iterations would regenerate the same fix, making the loop pointless. A very high temperature was also unsuitable as it would introduce excessive randomness into the repair, making the process unpredictable. Temperature 0.1 strikes a balance between the LLM exploring alternative fixes across iterations and keeping outputs close to the highest likelihood repair. This is validated in Section 4.3.2.
>
> **Q2 Response:**
>
> KLEE's driver code was initially generated using an LLM and then manually validated to ensure completeness (checking for missing symbolic variable declarations, proper memory allocation for pointer arguments, and correct constraint annotations). Any missing or incorrect harness logic was manually written and re-validated before running KLEE on each target project. This process was applied to ensure KLEE's performance was not underestimated due to poor driver quality.
>
> **Q3 Response:**
>
> We provide case studies from two complex subjects: lodepng (2,420 paths) and tulipindicators (517 paths) where SPARC outperforms KLEE (Table 1):
>
> 1. Lodepng:
>
> &nbsp;&nbsp;&nbsp;&nbsp;**a.** CRC32/Adler32 checksum verification: KLEE cannot solve complex math hashes to create valid files. SPARC uses LLM knowledge to build correct test files that pass these checks.
>
> &nbsp;&nbsp;&nbsp;&nbsp;**b.** Huffman decoding path explosion: The decompression loop creates O(6^N) symbolic branches per iteration, exhausting KLEE's state space, whereas SPARC generates path-targeted tests for specific decoding scenarios independently.
>
> 2. Tulipindicators:
>
> &nbsp;&nbsp;&nbsp;&nbsp;**a.** Floating-point black boxes: Transcendental functions (sqrt, log, sin, cos) are opaque external calls that KLEE must assign a concrete value to, losing all downstream branch exploration; SPARC reasons about their mathematical semantics to craft inputs exercising unique branches.
>
> &nbsp;&nbsp;&nbsp;&nbsp;**b.** Floating-point comparison branching: KLEE's SMT solvers (STP lacks FP theory entirely) cannot efficiently solve symbolic IEEE 754 comparisons, while SPARC generates concrete floating-point inputs guided by path-level reasoning about boundary conditions.
>
> &nbsp;&nbsp;&nbsp;&nbsp;**c.** Sequential dispatch with shared symbolic state: KLEE's single-driver design accumulates path constraints across all 104 functions on shared symbolic arrays, starving later functions of exploration budget; SPARC generates tests per function independently, avoiding cross-function state interference entirely.

---

### Official Review · Reviewer_DKLe · 2026-03-08

**Rating:** 3
**Confidence:** 3

**Review:**

### Strength

\+ The paper focuses on automated unit test generation for C projects, which is a timely and significant topic.

\+ The proposed framework is clearly structured and easy to follow.

\+ The reported experimental results are positive, demonstrating the effectiveness of SPARC on the evaluated subjects.


### Weakness

\- The novelty claim may be somewhat unclear.

\- The paper lacks important ablation studies for several components.

\- The scalability of SPARC for large projects is not very convincing.


### Comments for the authors

I really appreciate the authors’ efforts in proposing the SPARC framework for automated C unit test generation. The structured pipeline and the reported results are impressive. However, I have some concerns regarding the novelty claim, the lack of ablation studies, and the scalability of the framework.

First, the authors claim that SPARC is specifically designed for C projects, but from the structure of the framework, the special design for C is not very clear. From the introduction, it seems that SPARC is just overcoming the common issues in vanilla prompting and evaluated on C projects, but it did not naturally target the unique challenges of unit test generation for C projects. I would like to see more details on how SPARC is tailored for C projects and what specific challenges it addresses that are unique to C.

Second, the paper lacks important ablation studies for certain components of the SPARC framework. For example, SPARC utilizes an LLM to produce a semantic description of the function, and the descriptions are then used in the structural metadata. Why should the metadata be designed as in Section 3.3.1?  How much does the LLM-generated description contribute to the overall performance of SPARC? Similarly, the paper should also provide ablation studies for the main components of the framework to better understand their individual contributions. By the way, six RQs seem to be too many for this paper, and I suggest the author adjust the experiments to retain the most important ones.

Third, I am concerned about the scalability of SPARC.  Although the authors have studied SPARC on a project that has around 8K lines, it is still far from the practical C projects. Considering that the authors have analyzed the cost of SPARC on the studied projects and found that its token cost grows roughly quadratically with the number of control-flow paths, I wonder whether SPARC is suitable for real-world large-scale C projects and whether it would bring unbearable overhead in practice. This raises the concern that SPARC may not be efficient in achieving comprehensive test coverage for C projects, despite this being one of the main motivations stated in the introduction.

Lastly, I have some questions about the experiment results presentation. For example, there is no explanation for the shadowed column and the bold numbers. Do the bold numbers indicate the best performance among all three studied approaches or just among SPARC and DS? Also, why are the test cases generated by KLEE excluded from the mutation score analysis?

**Summary:**

This paper presents SPARC, a scenario-based framework for automated C unit test generation, which decomposes each function into per-path scenarios through four stages. This structured pipeline addresses common issues in vanilla prompting, including hallucinated dependencies, low branch coverage, and lack of traceability. The authors conduct evaluations on 59 real-world and algorithmic subjects, and the results indicate that SPARC outperforms the vanilla prompt generation baseline by 31.36% in line coverage, 26.01% in branch coverage, and 20.78% in mutation score, while matching or exceeding KLEE on complex subjects. Additionally, the framework retains 94.3% of generated tests through iterative repair and produces tests with higher readability, correctness, completeness, and maintainability.

---

> ### Author Response · Authors · 2026-03-19
>
> **Addressing the comments:**
>
> We thank the reviewer for the constructive feedback. We address your specific concerns below:
>
> **Regarding C-specificity:**
>
> We want to emphasize that we developed SPARC for C's architectural requirements at every stage. Our pipeline utilizes Clang and Tree-sitter for statement-level CFGs (Section 3.3.1) and leverages AddressSanitizer with GCC's --wrap=malloc (Section 3.3.4) to exercise manual memory failure paths. These are irrelevant to managed languages (Java/Python). Moreover, it is built on the Unity C framework (Section 3.3.3), and is validated by our RQ3 findings (Section 4.3), which capture language-specific issues such as malloc counter errors, memory ownership confusion, and address truncation. Please note that while a language-agnostic feature is valuable, focusing on specific properties of a programming language also results in a more stable pipeline for that language. $\underline{\text{We}}$ $\underline{\text{plan}}$ $\underline{\text{to}}$ $\underline{\text{use}}$ $\underline{\text{SPARC}}$ $\underline{\text{in}}$ $\underline{\text{a}}$ $\underline{\text{bigger}}$ $\underline{\text{pipeline}}$ $\underline{\text{to}}$ $\underline{\text{enhance}}$ $\underline{\text{validation}}$ $\underline{\text{of}}$ $\underline{\text{repository-level}}$ $\underline{\text{C-to-Rust}}$ $\underline{\text{translations}}$.
>
> **Regarding the ablation studies:**
>
> $\underline{\text{We}}$ $\underline{\text{designed}}$ $\underline{\text{our}}$ $\underline{\text{evaluation}}$ $\underline{\text{so}}$ $\underline{\text{that}}$ $\underline{\text{the}}$ $\underline{\text{ablation}}$ $\underline{\text{of}}$ $\underline{\text{SPARC's}}$ $\underline{\text{core}}$ $\underline{\text{components}}$ $\underline{\text{is}}$ $\underline{\text{baked}}$ $\underline{\text{into}}$ $\underline{\text{the}}$ $\underline{\text{RQs}}$. For instance, the DeepSeek baseline in RQ1 (Table 1) serves as a "full-system" ablation by removing the CFG-guided path enumeration and operation map (Section 4.1.2). To ensure these gains were not just the result of a powerful model, RQ6 isolates the pipeline's architecture by swapping out different LLM backbones. The results confirm that the architecture itself and not the specific model is the primary driver of quality. Finally, RQ2 demonstrates SPARC covers more unique control-flow paths than the baseline (Section 4.2.2), and even when coverage is equal, SPARC's scenario-guided oracles yield a higher mutation score.
>
> **Regarding the number of RQs:**
>
> Each of the six RQs targets a specific, necessary dimension of the system: effectiveness, validity, failure causes, perceived quality, cost, and portability. We believe that removing any of these would leave important questions unanswered.
>
> **Regarding scalability:**
>
> $\underline{\text{We want to clarify that SPARC operates at a per-function level instead of the full project (although it does account for intra- and inter- procedural dependencies between functions in the execution path)}}$. The pre-processing stage extracts each target function into an independent source file (Section 3.3.1). The quadratic cost reported in Section 4.5.1 is with respect to the number of control-flow paths per project, and not the project size (lines of code). Since test generation calls are independent (Section 3.3.3), adding more functions to a project scales linearly. The average token cost per test is quite practical shown in Section 4.5.2. Furthermore, as our variance analysis in Section 4.5.1 demonstrates, cost is significantly influenced by the structural complexity of the projects rather than path count alone.
>
> **Regarding table formatting and KLEE mutation scores:**
>
> We presented Table 1 in such a way that the $\underline{\text{bold}}$ $\underline{\text{numbers}}$ $\underline{\text{indicate}}$ $\underline{\text{the}}$ $\underline{\text{best}}$ $\underline{\text{performance}}$ $\underline{\text{between}}$ $\underline{\text{SPARC}}$ $\underline{\text{and}}$ $\underline{\text{DS}}$ (both LLM-based approaches) per metric. The shaded columns represent KLEE, visually separated as it belongs to a fundamentally different tool category (symbolic execution). We excluded KLEE from mutation score analysis because it only generates concrete test inputs and not complete test cases with assertions. Since mutation killing depends on assertion quality, computing scores for KLEE would reflect manual oracle-writing choices rather than KLEE's own generation capability.

---

### Official Review · Reviewer_6PHZ · 2026-03-08

**Rating:** 2
**Confidence:** 4

**Review:**

## Pros
+ Automated high-quality C unit test generation is a practically important problem
+ The scenario-based decomposition that aligns test generation with program structure is a reasonable approach.

## Cons
- Soundness concerns: the design lacks important details on loop handling during path extraction, does not verify path fidelity of generated tests, and does not guarantee the oracle correctness for generated assertions.
- Evaluation concerns: the benchmark is predominantly small-scale and missing comparison with general-purpose coding agents.



## Comments

Thanks for submitting your paper to AIware 2026. I appreciate the structured pipeline design that decomposes test generation into per-path scenarios grounded by Operation Maps, which is a reasonable approach to reducing LLM hallucination in C unit test generation. However, I have two main concerns:

### Soundness of the Design

First, the paper claims to enumerate "all feasible execution paths", but does not explain how loops are handled. In the presence of loops, the number of execution paths is theoretically infinite. The handling of loops is crucial for the soundness of the generated tests, but the details are missing.

Second, SPARC never verifies whether a generated test actually exercises the intended path $\pi$. The validation loop in design only checks for compilation errors, runtime errors, and assertion outcomes. The paper's own RQ2 data confirms this: only 75% of tests follow a unique control-flow path, meaning 25% are path-level redundant. Besides, the paper does not discuss what happens when the enumerated execution paths are actually unreachable.

Third, the paper does not discuss the oracle problem. The iterative repair fixes failing assertions by modifying the expected value to match actual output, but this conflates incorrect assertions with genuine bug detection. The paper shows in RQ3 that the LLM sometimes defaults to trivial inputs and the repair agent simply fixes the assertion to match, yet this raises concerns about the correctness of the generated assertions, which affects the quality of the generated tests. Would the assertion be to relax or even the opposite of the intent?

### Evaluation Concerns

The benchmark is predominantly small-scale: the 51 TheAlgorithms/C projects are single-file algorithm implementations (many less than 300 lines). Additionally, the paper does not compare against general-purpose coding agents, which have strong capabilities for iterative code generation and repair. Given that modern coding agents can retrieve code context, generate code, and iteratively fix errors, it is questionable whether this domain-specific pipeline design offers meaningful advantages over these general-purpose alternatives.

**Summary:**

This paper presents SPARC, a scenario-based framework for automated C unit test generation. SPARC decomposes unit test generation into a structured 4-stage pipeline: (1) CFG construction and path extraction using a static analysis tool, (2) RAG-augmented Operation Map construction that retrieves and assigns validated helper functions available during the test generation, (3) per-path test synthesis where each execution path receives a dedicated LLM call constrained by the Operation Map, and (4) iterative validation process with compiler and AddressSanitizer feedback. The authors evaluate SPARC on 59 C projects (51 from TheAlgorithms/C and 8 from Rustine) and compare SPARC against vanilla prompting and conventional symbolic executor KLEE. Results show SPARC achieves 26% higher branch coverage than vanilla prompting while being comparable to KLEE.

---

> ### Author Response · Authors · 2026-03-19
>
> Thank you very much for your review. We address your specific concerns below:
>
> **1. Soundness of the Design**
>
> &nbsp;&nbsp;&nbsp;&nbsp;**a. Loop Handling:** We clarify that SPARC utilizes an $\underline{\text{acyclic}}$ $\underline{\text{path}}$ $\underline{\text{model}}$ $\underline{\text{that}}$ $\underline{\text{specifically}}$ $\underline{\text{targets}}$ $\underline{\text{all}}$ $\underline{\text{feasible}}$ $\underline{\text{acyclic}}$ $\underline{\text{execution}}$ $\underline{\text{paths}}$. Our path extractor employs a Depth-First Search (DFS) that does not revisit nodes within a single path.
>
> &nbsp;&nbsp;&nbsp;&nbsp;**b. Verifying Path Fidelity and Redundancy:** The reviewer notes that 75% of generated tests follow unique paths which is much higher than DS (Section 4.2.2). This suggests that providing the LLM with path constraints effectively steers execution. Moreover, among tests sharing paths, SPARC achieves higher mutation scores than the baseline (Table 1), and even at equal coverage, confirming that path-guided generation produces stronger oracles, not just more paths. $\underline{\text{Regarding}}$ $\underline{\text{unreachable}}$ $\underline{\text{paths,}}$ $\underline{\text{RQ3}}$ $\underline{\text{quantifies}}$ $\underline{\text{these}}$ $\underline{\text{as}}$ $\underline{\text{failures}}$. When the path extractor selects dead code, the test either fails generation or is filtered out during validation and excluded from the final suite.
>
> &nbsp;&nbsp;&nbsp;&nbsp;**c. The Oracle Problem and Correctness:** SPARC generates tests for known-correct programs, so matching assertions to actual output is standard practice in automated test generation. The repair loop (Section 4.3.2) processes full compiler and AddressSanitizer diagnostics, and only 1 of 37 runtime failures in Section 4.2.1 is an assertion mismatch. More importantly, mutation testing validates oracle quality independently: SPARC achieves 20.78% higher mutation scores than the baseline (Section 4.2.2, Table 1), and on qsort, where both approaches reach identical coverage, SPARC still kills 6% more mutants (87% vs. 81%). This gap is due to stronger assertions.
>
> **2. Evaluation Concerns**
>
> &nbsp;&nbsp;&nbsp;&nbsp;**a. Benchmark Scale:** SPARC operates at the per-function level rather than the project level. The pre-processing stage extracts each target function into an independent source file (Section 3.3.1, Figure 3), so project size (LOC) is not the relevant complexity metric. $\underline{\text{Since}}$ $\underline{\text{per-path}}$ $\underline{\text{test}}$ $\underline{\text{generation}}$ $\underline{\text{calls}}$ $\underline{\text{are}}$ $\underline{\text{independent}}$ $\underline{\text{(Section 3.3.3),}}$ $\underline{\text{adding}}$ $\underline{\text{more}}$ $\underline{\text{functions}}$ $\underline{\text{scales}}$ $\underline{\text{linearly}}$. Notably, on the largest subject (lodepng), KLEE's constraint solver begins to struggle while SPARC maintains strong coverage (Table 1).
>
> &nbsp;&nbsp;&nbsp;&nbsp;**b. Comparison against general-purpose coding agents:** Thanks for your suggestion. Generally speaking, $\underline{\text{every}}$ $\underline{\text{neuro-symbolic}}$ $\underline{\text{pipeline}}$ $\underline{\text{such}}$ $\underline{\text{as}}$ $\underline{\text{SPARC}}$ $\underline{\text{can}}$ $\underline{\text{be}}$ $\underline{\text{transformed}}$ $\underline{\text{to}}$ $\underline{\text{an}}$ $\underline{\text{agentic}}$ $\underline{\text{scaffold}}$ by writing interfaces to use the tooling in the neurosymbolic pipeline as tools for agents. $\underline{\text{Our}}$ $\underline{\text{decision}}$ $\underline{\text{to}}$ $\underline{\text{stick}}$ $\underline{\text{with}}$ $\underline{\text{a}}$ $\underline{\text{neuro-symbolic}}$ $\underline{\text{pipeline}}$ $\underline{\text{is}}$ $\underline{\text{two-fold}}$: first, the cost matters and we envision SPARC to be used as an important component in existing repository-level C-to-Rust translation pipelines. Such techniques largely rely on tests for validating translations, but at the same time, many real-world C projects lack tests with high coverage and quality assertions. Second, adhering to the workflow is essential in SPARC. However, agents may or may not adhere to the specified workflow through their chain of ReAct-based trajectories.